# Fitness benefits in fluoroquinolone-resistant *Salmonella* Typhi in the absence of antimicrobial pressure

Stephen Baker[1,2,3]*, Pham Thanh Duy[1], Tran Vu Thieu Nga[1], Tran Thi Ngoc Dung[1], Voong Vinh Phat[1], Tran Thuy Chau[1], A Keith Turner[4], Jeremy Farrar[1,2], Maciej F Boni[1,2]

[1]Oxford University Clinical Research Unit, Wellcome Trust Major Overseas Programme, Ho Chi Minh City, Vietnam; [2]Centre for Tropical Medicine, Nuffield Department of Clinical Medicine, University of Oxford, Oxford, United Kingdom; [3]The London School of Hygeine and Tropical Medicine, London, United Kingdom; [4]Wellcome Trust Genome Campus, The Wellcome Trust Sanger Institute, Hinxton, United Kingdom

**Abstract** Fluoroquinolones (FQ) are the recommended antimicrobial treatment for typhoid, a severe systemic infection caused by the bacterium *Salmonella enterica* serovar Typhi. FQ-resistance mutations in *S.* Typhi have become common, hindering treatment and control efforts. Using in vitro competition experiments, we assayed the fitness of eleven isogenic *S.* Typhi strains with resistance mutations in the FQ target genes, *gyrA* and *parC*. In the absence of antimicrobial pressure, 6 out of 11 mutants carried a selective advantage over the antimicrobial-sensitive parent strain, indicating that FQ resistance in *S.* Typhi is not typically associated with fitness costs. Double-mutants exhibited higher than expected fitness as a result of synergistic epistasis, signifying that epistasis may be a critical factor in the evolution and molecular epidemiology of *S.* Typhi. Our findings have important implications for the management of drug-resistant *S.* Typhi, suggesting that FQ-resistant strains would be naturally maintained even if fluoroquinolone use were reduced.

**\*For correspondence:** sbaker@oucru.org

## Introduction

The evolution of antimicrobial resistance in bacteria is driven by the pressure of sustained exposure to antimicrobials. Such strong selective pressure has stark short and long-term consequences, as evidenced by more than half a century of antimicrobial usage and resistance evolution (*Andersson and Hughes, 2010*). The development of specific antimicrobial resistance patterns within sentinel organisms has spawned the popular term 'super-bug' (*Foster, 2004*; *Anzaldi and Skaar, 2011*). This term is misleading in the sense that resistance to a specific antimicrobial typically confers a reduction in Darwinian fitness (a fitness cost) in the absence of the pressure induced by that antimicrobial (*Andersson and Levin, 1999*; *Andersson, 2003*). Nevertheless, bacterial mutations that appear to have a low fitness cost or no fitness cost are sometimes observed, and mutations that may induce a fitness benefit in the absence of antimicrobials have been described but are rare (*Macvanin et al., 2003*; *Enne et al., 2004*; *Luo et al., 2005*; *Kassen and Bataillon, 2006*; *Rozen et al., 2007*; *Marcusson et al., 2009*; *Bataillon et al., 2011*; *Miskinyte and Gordo, 2013*). Understanding the fitness effects of antimicrobial resistance evolution is crucial for controlling the spread of resistance, as the fitness cost induced by antimicrobial resistance is one of the few biological features of resistant organisms that can be leveraged against them.

Over the past 20 years, as a consequence of resistance to multiple first-line antimicrobials in the *Enterobacteriacea*, clinicians have become progressively reliant on the fluoroquinolones for treating

**eLife digest** The fluoroquinolones are a group of antimicrobials that are used to treat a variety of life-threatening bacterial infections, including typhoid fever. Before the introduction of antimicrobials, the mortality rate from typhoid fever was 10–20%. Prompt treatment with fluoroquinolones has reduced this to less than 1%, and has also decreased the severity of symptoms suffered by people with the disease.

Now, however, the usefulness of many antimicrobials, including the fluoroquinolones, is threatened by the evolution of antimicrobial resistance within the bacterial populations being treated. Drug resistance in bacteria typically arises through specific mutations, or following the acquisition of antimicrobial resistance genes from other bacteria. It is thought that the frequent use of antimicrobials in human and animal health puts selective pressure on bacterial populations, allowing bacterial strains with mutations or genes that confer antimicrobial resistance to survive, while bacterial strains that are sensitive to the antimicrobials die out.

At first it was thought that specific mutations conferring antimicrobial resistance came at a fitness cost, which would mean that such mutations would be rare in the absence of antimicrobials. Now, based on research into typhoid fever, Baker et al. describe a system in which the majority of evolutionary routes to drug resistance are marked by significant fitness benefits, even in the absence of antimicrobial exposure.

Typhoid is caused by a bacterial pathogen known as *Salmonella* Typhi, and mutations in two genes—*gyrA* and *parC*—result in resistance to fluoroquinolones. Baker et al. show that mutations in these genes confer a measurable fitness advantage over strains without these mutations, even in the absence of exposure to fluoroquinolones. Moreover, strains with two mutations in one of these genes exhibited a higher than predicted fitness, suggesting that there is a synergistic interaction between the two mutations. This work challenges the dogma that antimicrobial resistant organisms have a fitness disadvantage in the absence of antimicrobials, and suggests that increasing resistance to the fluoroquinolones is not solely driven by excessive use of this important group of drugs.

infections caused by this Gram-negative bacterial family (*Parry, 2004*; *Vinh et al., 2011*). The increasing usage and dependency on the fluoroquinolones has coincided with resistance to fluoroquinolones becoming common within the *Enterobacteriacea* (*Le et al., 2009*; *Butler, 2011*). This change is highly pertinent for diseases like typhoid, a severe invasive infection caused by the bacterium *Salmonella enterica* serovar Typhi (*Parry et al., 2002*), as antimicrobial therapy is essential for treatment. Typhoid remains common in many developing countries, and increasing minimum inhibitory concentrations (MICs) to fluoroquinolones, such as ciprofloxacin and ofloxacin, correlate with increased fever clearance times and treatment failure (*Parry et al., 2011*). The elevated MICs to fluoroquinolones observed in *S.* Typhi are a consequence of mutations in the DNA gyrase gene, *gyrA*, and secondary mutations in the topoisomerase gene, *parC* (*Chau et al., 2007*; *Parry et al., 2010*). Evidence of strong selection for these mutations can be observed within the phylogenetic structure of *S.* Typhi, as several different *gyrA* mutations have arisen independently in multiple lineages (*Roumagnac et al., 2006*; *Holt et al., 2008*).

To determine how fluoroquinolone-resistance evolution in *S.* Typhi impinges on the relative fitness of this pathogen, we directly competed strains in a series of controlled experiments, using isogenic strains to isolate the fitness effects of specific mutations, and employing pyrosequencing for precise measurement of allele frequencies. In classical competition assays (*Lenski, 1991*; *Lenski et al., 1998*), antimicrobial-susceptible and antimicrobial-resistant organisms are competed over many generations, and the frequencies of resistant and sensitive strains are compared at various time points. The relative fitness of the resistant strain to the sensitive strain can be calculated from the population trajectories observed in the experiment (*Macvanin et al., 2003*; *Enne et al., 2005*; *Gagneux et al., 2006*; *Balsalobre and de la Campa, 2008*). The choice of bacterial strains is critical in performing this type of competitive growth assay. Competing genetically unrelated clinical isolates (*Laurent et al., 2001*; *Wichelhaus et al., 2002*), or strains that are otherwise imperfectly isogenic, may make it difficult to isolate the effects of a single mutation (*Enne et al., 2005*; *Komp Lindgren et al., 2005*; *Gagneux et al., 2006*; *MacLean and Buckling, 2009*; *O'Regan et al., 2010*). Furthermore, bacterial enumeration and

selective culturing after serial dilutions are typically used to calculate population sizes (*Laurent et al., 2001*; *Wichelhaus et al., 2002*; *Macvanin et al., 2003*; *Gagneux et al., 2006*; *Rozen et al., 2007*; *Balsalobre and de la Campa, 2008*; *MacLean and Buckling, 2009*; *Randall et al., 2008*), and these methodologies can be affected by experimental variation or spontaneous mutations in the target gene(s) as a consequence of exposure to low levels of antimicrobial. To overcome these limitations, we performed competitive growth experiments with isogenic strains containing single and multiple mutations in fluoroquinolone target genes, and we used pyrosequencing to assay allele frequency, avoiding exposing the organisms to antimicrobials.

As *gyrA* mutations and susceptibility to fluoroquinolones appear to be under strong selective pressure in *S.* Typhi, we assessed the biological fitness of *S.* Typhi strains with a relevant complement of mutations in the *gyrA* and *parC* genes. We demonstrate that a large proportion of clinically and epidemiologically relevant *gyrA* mutations induce significant fitness benefits in *S.* Typhi in the absence of antimicrobial pressure. We show that strong epistatic interactions between loci in the *gyrA* and *parC* genes of *S.* Typhi confer additional significant selective advantages that may be responsible for driving the evolution and current regional expansion of *S.* Typhi in the developing countries.

## Results

### MICs of *S.* Typhi mutants

We constructed 12 individual *S.* Typhi mutants, seven of which have been isolated clinically, by introducing one or more single mutations into the *gyrA* and *parC* genes of a host *S.* Typhi strain by allelic exchange (*Turner et al., 2006*). A strain description and the minimum inhibitory concentrations (MICs) against nalidixic acid and a range of fluoroquinolones are shown in *Table 1*. The seven mutants with naturally occurring equivalents all demonstrated significant increases in MICs over the parent strain with all tested antimicrobials. The increases in MICs were comparable to those observed in clinical isolates with the corresponding natural mutations (*Chau et al., 2007*; *Parry et al., 2010*). Double mutants exhibited greater MICs than single mutants and the triple mutant exhibited the highest MIC to all tested fluoroquinolones (*Parry et al., 2010*). The four strains without a naturally occurring counterpart also demonstrated higher MICs than the parent strain, and the control strain—a strain containing a mutation in the defunct *aroC* gene—demonstrated no significant difference from the parent *S.* Typhi strain (*Table 1*).

### Allele frequency calculation with pyrosequencing

We assessed the selective advantages/disadvantages of the 12 mutants relative to the parent *S.* Typhi strain through competitive growth experiments. Because (i) our ability to differentiate strains by bacterial culture on selective media was limited as a consequence of low MICs, and (ii) serial dilution and colony enumeration, with and without antimicrobial, generated extensive experimental variation between replicates (*Figure 1*), we developed three pyrosequencing assays to discriminate the parent *S.* Typhi strain from the single/double nucleotide variations in *gyrA*, the single mutation in *parC*, and the single mutation in *aroC*.

To validate the pyrosequencing methodology, we combined known concentrations of bacterial cultures of the parent *S.* Typhi strain and the S83F mutant and compared bacterial dilution and enumeration against allelic frequency detection using pyrosequencing. Allele frequencies measured by bacterial enumeration and pyrosequencing demonstrated a strong linear relationship ($r^2 = 0.90$, *Figure 1*), with measurement by pyrosequencing exhibiting less variation among replicates than classical culture and enumeration ($p < 4 \times 10^{-4}$ for all nine starting dilutions between 0.1 and 0.9; $6.4 < F_{17,17} < 38.1$). These data indicate our pyrosequencing assay is an accurate and highly reproducible method for determining the specific allele frequencies in mixed bacterial suspensions over a range on concentrations.

### Selection coefficients of *S.* Typhi mutants

We combined the parent *S.* Typhi with each of the mutants described in *Table 1* and performed 12 competitive growth experiments (five replicates each), lasting between 152 and 158 generations (15 days). Selection coefficients (*s*) for the 12 mutants were defined as per-generation percentage reductions or increases in fitness, relative to the parent (wild-type) strain. For example, if a strain has an estimated $\hat{s} = -.03$, this means that on average this strain will produce 3% fewer surviving offspring than the parent strain, in one bacterial generation. The fitness coefficient (*w*) is defined as $w = 1 + s$. Maximum likelihood estimates (MLE) of the selection coefficients of two mutants (D87N, S83Y) were

**Table 1.** S. Typhi mutants constructed for this study

| S. Typhi strain | Genotype | Minimum inhibitory concentrations (µg/ml) | | | | | |
|---|---|---|---|---|---|---|---|
| | | Nalidixic acid | Norfloxacin | Ofloxacin | Ciprofloxacin | Gatifloxacin | Levofloxacin |
| Parent BRD948 | ΔaroA, ΔaroC, ΔhtrA | 1.5 | 0.064 | 0.047 | 0.008 | 0.008 | 0.012 |
| DPT001 | SNP in ΔaroC (codon 10) | 1.5 | 0.064 | 0.047 | 0.008 | 0.008 | 0.012 |
| S83Y | SNP in gyrA (codon 83) | 256 | 0.5 | 0.25 | 0.125 | 0.125 | 0.125 |
| S83F | SNP in gyrA (codon 83) | 256 | 0.75 | 0.38 | 0.125 | 0.125 | 0.125 |
| D87A | SNP in gyrA (codon 87) | 48 | 0.75 | 0.19 | 0.094 | 0.064 | 0.064 |
| D87N | SNP in gyrA (codon 87) | 48 | 0.75 | 0.25 | 0.125 | 0.125 | 0.125 |
| D87G | SNP in gyrA (codon 87) | 48 | 0.75 | 0.25 | 0.125 | 0.125 | 0.25 |
| S80I | SNP in parC (codon 80) | 3 | 0.19 | 0.047 | 0.016 | 0.016 | 0.016 |
| D87G-S80I | SNP in gyrA (codon 87) and SNP in parC (codon 80) | 256 | 1 | 0.25 | 0.125 | 0.094 | 0.094 |
| S83F-D87G | 2 SNPs in gyrA (codons 83 and 87) | 256 | 1 | 0.38 | 0.19 | 0.25 | 0.25 |
| S83F-D87A | 2 SNPs in gyrA (codons 83 and 87) | 192 | 1.5 | 0.38 | 0.25 | 0.38 | 0.25 |
| S83F-D87N | 2 SNPs in gyrA (codons 83 and 87) | 64 | 0.75 | 0.38 | 0.19 | 0.19 | 0.19 |
| S83F-D87G-S80I | 2 SNP in gyrA (codons 83 & 87) and SNP in parC (codon 80) | 256 | 24 | 16 | 8 | 2 | 3 |

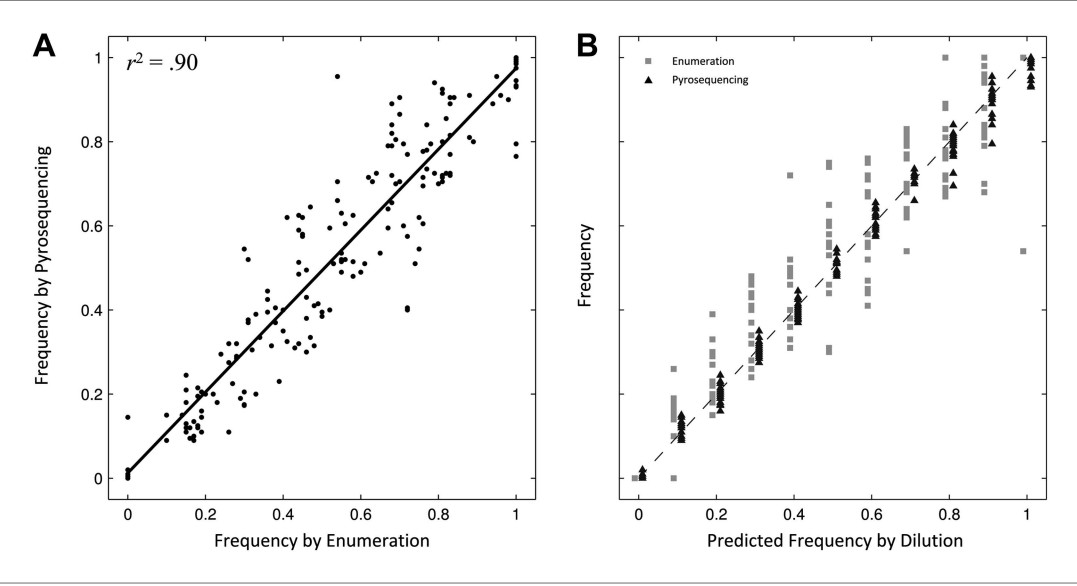

**Figure 1**. Comparing two methods for calculation of allele frequencies. (**A**) Pyrosequencing-measured allele frequencies (y-axis) of a range of S83F/parent strain dilutions plotted against enumeration-measured frequencies (x-axis) (n = 198). A linear regression between the two variables (solid black line) explains 90% of the variation in the relationship between these two measurements. (**B**) The same 198 data points (y-axis) are shown plotted against the original bacterial dilution ratio (x-axis). The broken line is the diagonal highlighting where predicted frequency and measured frequency would be identical. 18 measurement replicates were performed for each predicted frequency of S83F from 0.0 to 1.0.

not statistically different from zero (exhibiting neutrality), while selection coefficients for three mutants (D87A, D87G, and the triple mutant S83F-D87G-S80I) were slightly negative indicating that these mutants carried fitness costs (*Figure 2*). The largest fitness cost was observed in the triple mutant that demonstrated a 1% reduction in fitness per bacterial generation (MLE $\hat{s} = -.010$). The remaining six mutants had statistically significant positive selection coefficients, ranging from 0.8% to 7.4% per bacterial generation. The six high-fitness mutants were two strains with a single mutation (S83F and S80I) and all four double mutants: S83F-D87A, S83F-D87G, S83F-D87N, and D87G-S80I. The single mutant with the highest selection coefficient was S83F (MLE $\hat{s} = +.013$) and the double S83F-D87N mutant displayed the highest selection coefficient of all tested strains (MLE $\hat{s} = +.074$).

To exclude the possibility of compensatory mutations arising during the experiments and affecting bacterial fitness (*Bataillon et al., 2011*; *Sousa et al., 2012*), the fitness coefficients for the first 5 days of each 15-day assay were recomputed (*Figure 3*). The expected pattern under a scenario of compensatory evolution would be slow evolution in the early phases of the competition experiment with more rapid selection in the later stages. With the exception of the S83Y mutant, none of the competition experiments exhibited this pattern; and for S83Y, both fitness coefficients had confidence intervals that included zero. Therefore, unless compensatory mutations occurred and became fixed in the very early stages of the competition experiments, compensation did not have an effect on our estimation of fitness coefficients.

## Epistasis within gyrA and between gyrA and parC mutations

All of the double mutants demonstrated highly increased fitness over the *S*. Typhi parent strain and all single mutants, suggesting that epistatic interactions among resistance loci may play a role in determining strain fitness. Defining fitness for the double mutants as $w_{ij} = (1 + s_i)(1 + s_j) + \varepsilon_{ij}$, we obtained MLEs for the epistasis parameters ($\varepsilon_{ij}$), all of which were statistically different from zero and positive, indicating that the combined fitness effect of two *gyrA* mutations (or in one case, a *gyrA* and a *parC* mutation) was greater than that predicted under a scenario of multiplicative non-epistasis (*Figure 4*). Notably, all three single mutations at codon 87 in the *gyrA* gene were selectively neutral or nearly-neutral unless in combination with the S83F mutation or the S80I *parC* mutation. The S83F and D87N

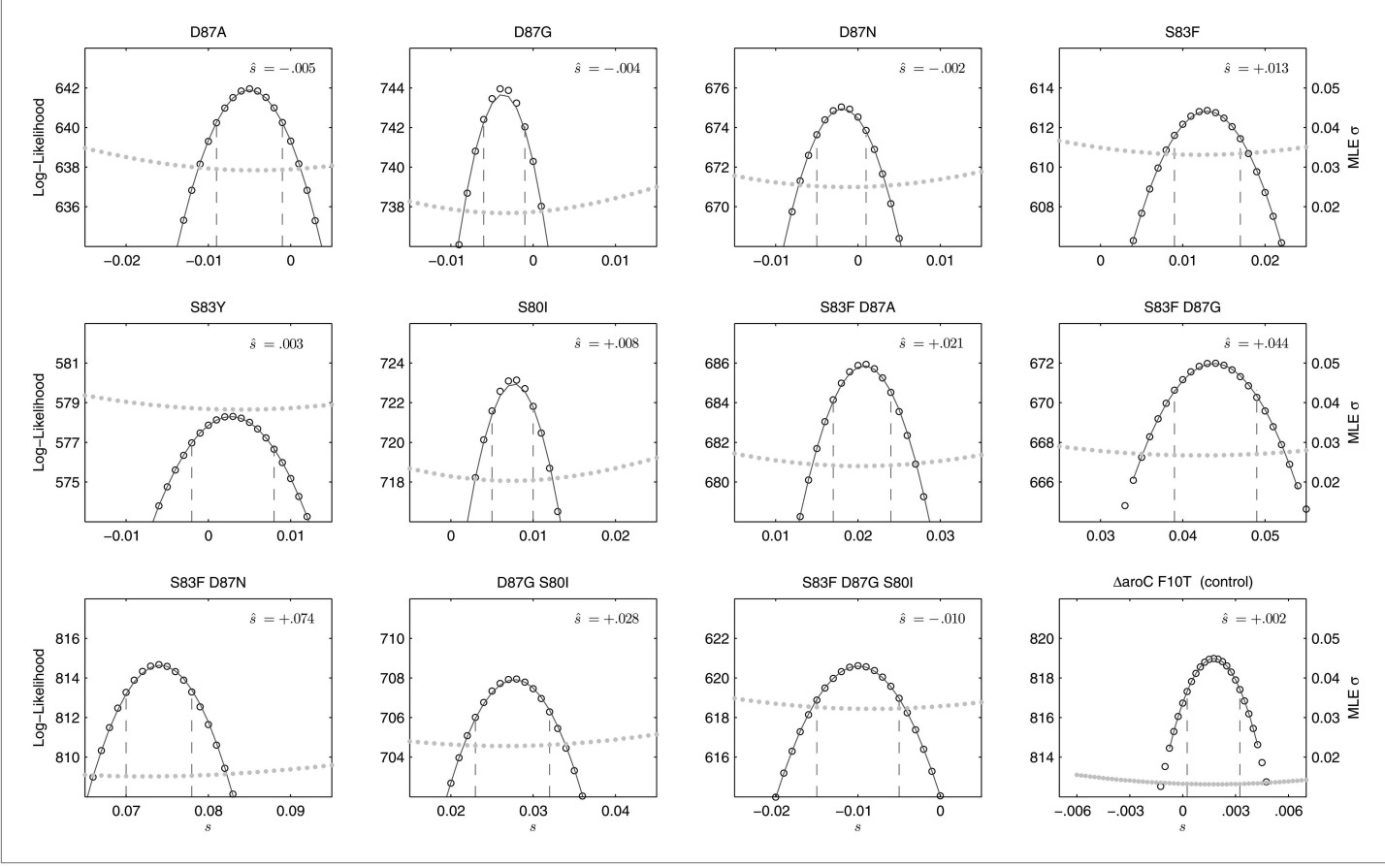

**Figure 2**. Likelihood profiles for the selection coefficients from 12 competition experiments. Data generated by competing 12 *S.* Typhi mutants (labeled at the top of each panel) against the parent *S.* Typhi strain over approximately 150 generations. Open circles correspond to likelihood values over the entirety of the experiment (primary y-axis); the filled gray circles correspond to the maximum likelihood estimates (MLE) for the variance parameter σ (secondary y-axis), describing the 24-hourly variance in both process and measurement. The MLE selection coefficient ($\hat{s}$) is shown in the top right of each panel. Vertical dashed lines demark the 95% confidence intervals for the MLE $\hat{s}$. Note the compressed x-axis scale in the bottom-right panel.

mutations demonstrated the greatest degree of synergistic epistasis, with a 6.6% increase in fitness resulting from the epistatic interaction. The relationship among the various *S.* Typhi mutants evaluated in this study, their MICs, selection coefficients, and epistatic interactions are summarized in **Figure 5**.

We also calculated the epistasis coefficient for the S83F-D87G-S80I triple mutant relative to the parent strain. The epistasis coefficient for this triple mutant was negative at −2.6% (**Figure 6**). However, this epistasis calculation assumes that a single interaction coefficient modulates the interaction of fitness effects among these three mutations. The actual interactions in nature would occur as single mutations emerging on the background of a double mutant. As two of the relevant double mutants were generated for our experiments, we were able to calculate the epistasis coefficient of S80I emerging onto a background of S83F-D87G and that of S83F emerging onto a background of D87G-S80I. The S83F-D87G mutant has been observed clinically and its interaction with the S80I mutation is antagonistic (MLE $\hat{\varepsilon} = −.061$); this is the most likely pathway to the triple mutant. The interaction between S83F and the laboratory-generated double mutant D87G-S80I is similarly negative (MLE $\hat{\varepsilon} = −.051$). Likelihood profiles for the three possible epistatic interactions leading to the triple mutant are shown in **Figure 6**.

## Discussion

In exploring the Darwinian fitness of *S.* Typhi strains containing a range of clinically and epidemiologically relevant fluoroquinolone-resistance mutations, we endeavored to address some of the drawbacks that may hinder the calculation of accurate fitness coefficients in other bacterial systems. To this end, we generated mutants from the same parent through controlled site-directed mutagenesis, validating all

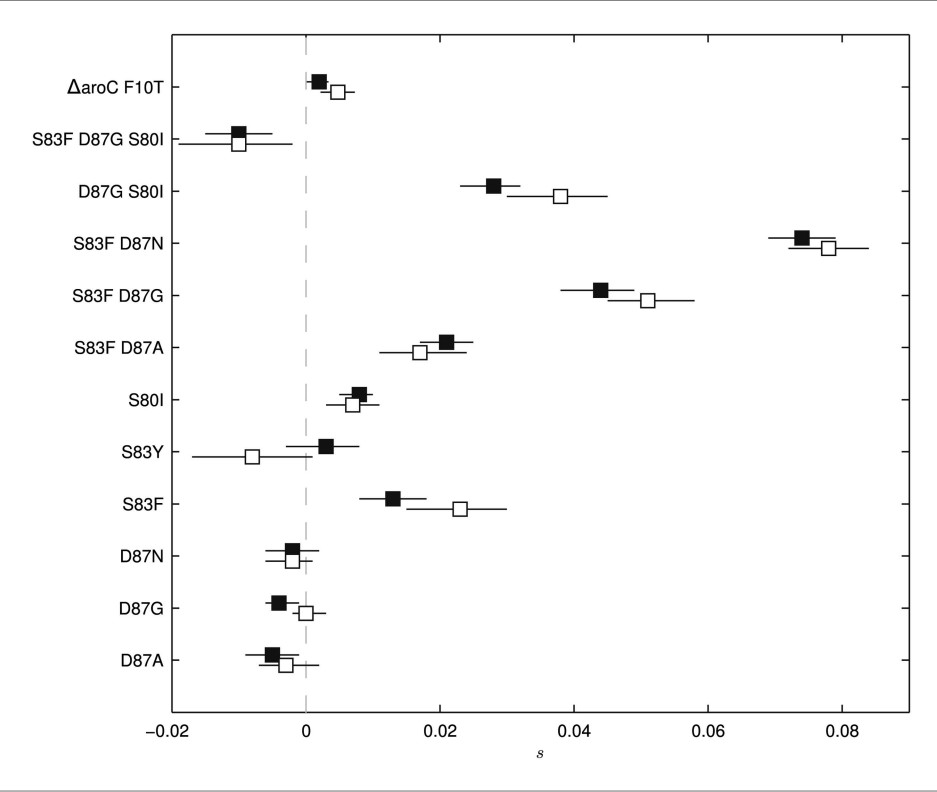

**Figure 3**. Fitness coefficients computed from 5 and 15 days of bacterial competition. Black boxes show fitness coefficients computed across the entire 15-day competition. White boxes show fitness coefficients computed from the first 5 days only. The ΔaroC F10T mutation is that of the control strain. Horizontal lines are 95% confidence intervals. In a situation of compensatory evolution, we would expect to see the white box to the left of the black box.

nucleotide substitutions, in order to avoid the use of clinical isolates from unspecified genetic backgrounds. Additionally, we used pyrosequencing to determine allele frequencies in order to minimize assay fluctuation, bacterial enumeration error, and other biases that may result from exposing *S.* Typhi to antimicrobials during the experimental procedure. This pyrosequencing approach permitted a greater degree of precision than serial dilution and colony counting, and it can be readily applied to a range of experimental systems without the requirement of a phenotypic marker.

Our experimental results run contrary to the dogma that antimicrobial-resistant organisms exhibit a selective disadvantage in the absence of antimicrobials. Notable exceptions to this rule include studies in quinolone-resistant *Neisseria gonorrhoeae* (**Kunz et al., 2012**), sulphanomide-resistant *E. coli* (**Kassen and Bataillion, 2006**), and fluoroquinolone-resistant *Campylobacter jejuni* (**Luo et al., 2005**) *Streptococcus pneumoniae* (**Rozen et al., 2007**) and *E. coli* (**Marcusson et al., 2009**), all of which described individual genotypes with both increased MIC and higher intrinsic fitness. Studies on *Pseudomonas fluorescens* (**Kassen and Bataillon, 2006**; **Bataillon et al., 2011**) have shown that a small fraction of laboratory-generated mutants can harbor both drug resistance and fitness benefits in the absence of antimicrobial pressure. Most recently, an investigation by **Miskinyte and Gordo (2013)** suggested that *E. coli* with antimicrobial resistance mutations have measurable fitness benefits inside macrophages. Our work adds to these finding by demonstrating a pathogen-drug combination in which the majority of resistant genotypes are associated with dramatic fitness benefits in the absence of antimicrobial pressure. Our findings have implications for the control of typhoid as enhanced fitness in the absence of antimicrobial pressure eliminates the option of prudent antimicrobial use as a public health strategy. Additionally, our results show that combinations of these mutations exhibit higher Darwinian fitness than would be expected under a scenario of independent fitness effects, showing that higher fitness in *gyrA* and *parC* mutants arises from synergistic epistasis. It has been shown previously

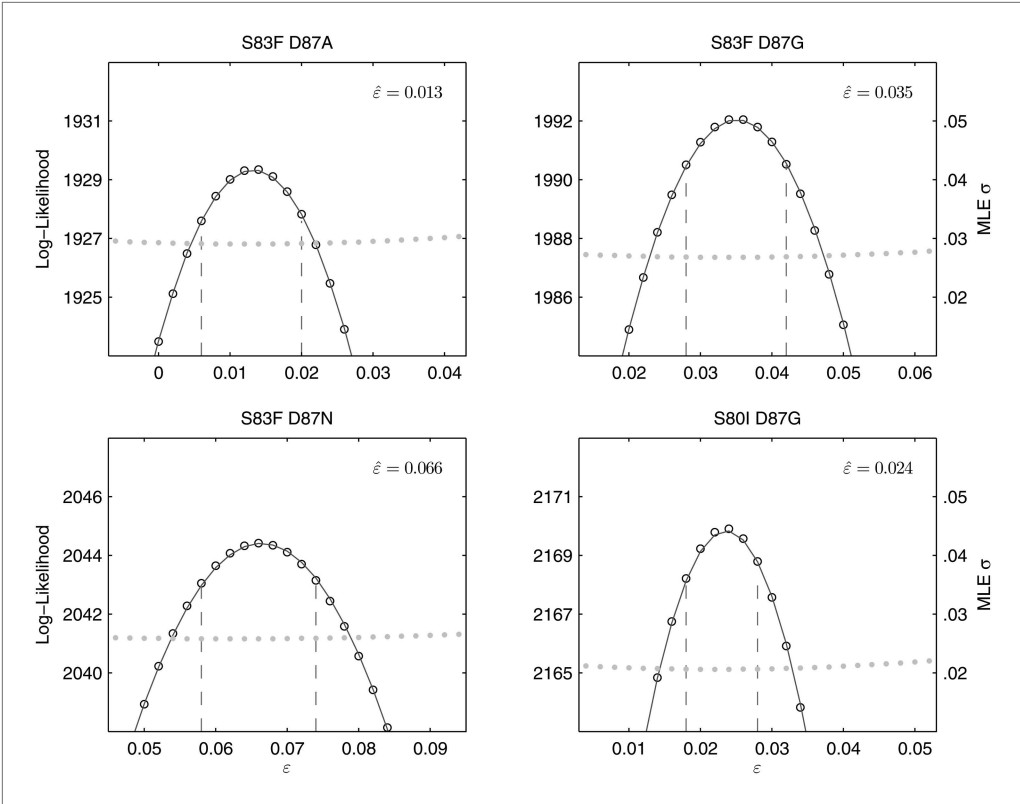

**Figure 4**. Likelihood profiles for the epistasis coefficient ($\hat{\varepsilon}$) from the four double mutant competition experiments. Open circles correspond to likelihood values; the filled gray circles correspond to the maximum likelihood estimate (MLE) for the variance parameter σ, describing the 24-hourly variance in both process and measurement. The MLE epistasis coefficient $\hat{\varepsilon}$ is shown in the top right of each panel. Vertical dashed lines demark the 95% confidence intervals for the MLE $\hat{\varepsilon}$.

that genome-wide epistatic interactions may aid the development of multidrug resistance (*Trindade et al., 2009*; *Silva et al., 2011*), but the described mutations in these studies are not found in genes that are directly related to antimicrobial resistance.

The potential impact of our results must be viewed in the context of certain limitations of *S.* Typhi experimental systems and our current understanding of *S.* Typhi biology. *S.* Typhi is a human-restricted facultative intracellular organism, and the selection coefficients that we observed using an *S.* Typhi *aro* mutant strain in an in vitro media-derived competitive growth assay system may not be replicated in other experimental systems. The selected parent (*S.* Typhi BRD948) is a well-studied non-invasive laboratory strain and was selected to avoid potential biological safety issues attributed to introducing known antimicrobial resistance mutations into an invasive human pathogen. If future experimentation were to be performed with invasive *S.* Typhi strains, this would provide better evidence that the in vitro fitness effects we observed would be similar to those observed in a natural epidemiological setting. However, there is currently no in vivo or in vitro experimental system outside humans or primates that accurately mimics an *S.* Typhi infection cycle.

The options for future experimentation regarding the fitness of fluoroquinolone-resistance mutations in *Salmonella* are in vitro and ex vivo cellular systems or the available murine models. All options have their own inherent limitations with respect to typhoid fever in humans, but they also have the potential to provide informative data. *S.* Typhi invades intestinal epithelial cells (M cells) and uses the macrophage as a vehicle for systemic dissemination; therefore an epithelial cell (*Bishop et al., 2008*) or macrophage (*Segura et al., 2004*) invasion/replication assay may provide appropriate fitness measurements for *S.* Typhi mutants in an intracellular system. There are physiological limitations with such cellular systems such as uptake, antimicrobial exposure (gentamycin), and cellular replication that may hinder experimental reproducibility. A more suitable approach may be the classical *S.* Typhimurium

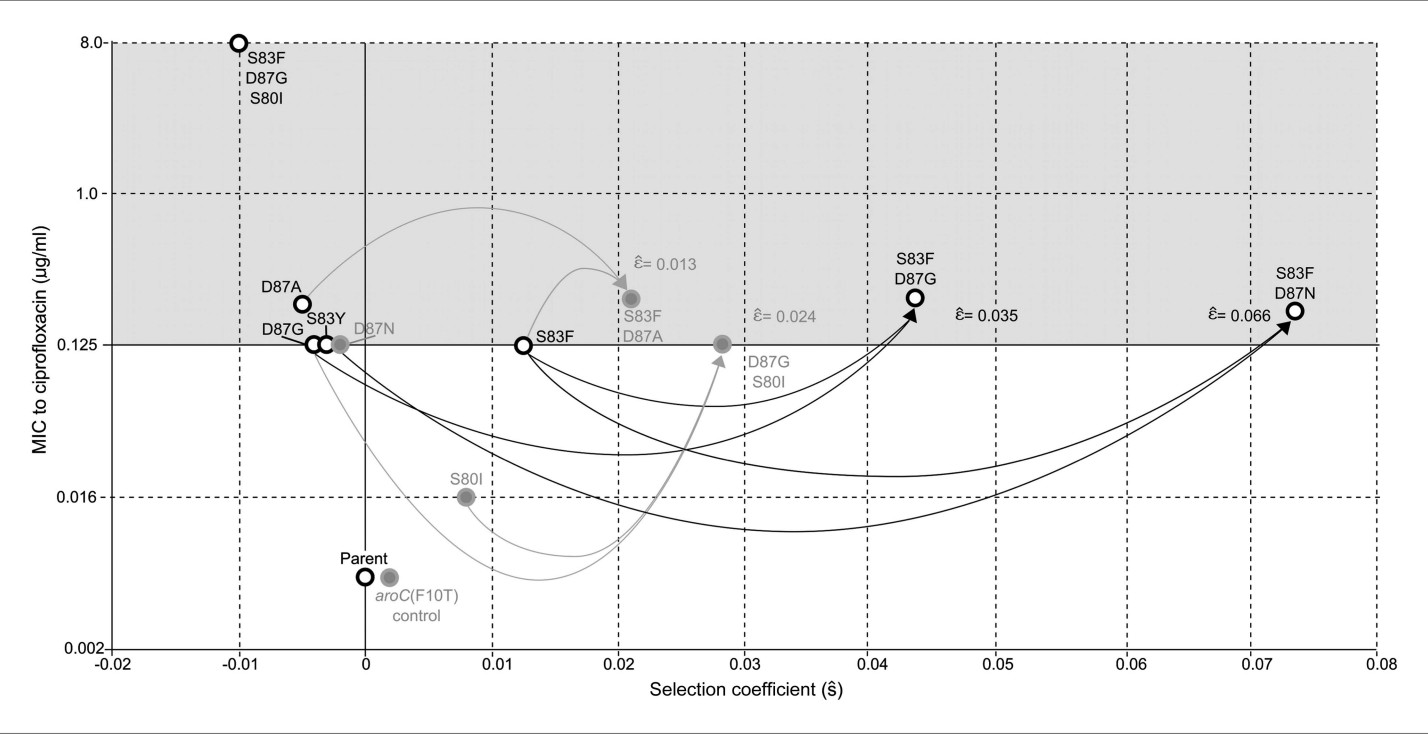

**Figure 5**. Relationships among MICs, selection coefficients and epistasis parameters of *S*. Typhi mutants. Diagram depicts the interactions among MLE selection coefficients ($\hat{s}$) (x-axis), MICs to ciprofloxacin (y-axis), and MLE epistasis coefficients $\hat{\varepsilon}$. Black circles denote *S*. Typhi strains that have been isolated clinically, while gray circles denote *S*. Typhi strains that have not been isolated clinically. Lines correspond to epistatic interactions of the four double mutants, two of which have been isolated clinically (black lines and $\hat{\varepsilon}$ value) and two of which have not (gray lines and $\hat{\varepsilon}$ value). The grayed upper half of graph highlights the current MIC breakpoint indicative of resistance and increasing risk of treatment failure (>0.125 µg/ml).

challenge model. Direct competition of attenuated *S*. Typhimurium mutants in a mouse model is well described, and *gyrA* mutations in *S*. Typhimurium could be generated and compared using in vivo competition assays. One potential limitation of this animal model in determining accurate selection coefficients is the brief duration of infection, which results in a small number of bacterial generations over which to observe fitness differences. The importance of replicating our finding in other systems is highlighted by two previous studies on *S*. Typhimurium that assessed fitness effects associated with mutations in *gyrA*, one of which showed a selective disadvantage of fluoroquinolone-resistant strains in the gut of chickens (*Giraud et al., 2003*), and a second of which demonstrated a reduction in invasion measured in immortalized epithelial cells (*Fàbrega et al., 2009*). The strains in both studies were generated by serial passage to select for resistant isolates, which may have induced a range of uncontrolled secondary mechanisms.

An additional caveat in considering a suitable experimental model relates to our current understanding of *S*. Typhi epidemiology. Fitness advantages in a transmission context may not correspond to those observed in vitro, ex vivo, or in animal models. Genotype-by-environment interactions have been observed to have significant effects on bacterial fitness measurements (*Paulander et al., 2009*; *Miskinyte and Gordo, 2013*), and it is not clear which environments play the largest real-life role in modulating the epidemiological fitness of different *S*. Typhi strains. For example, individuals that carry *S*. Typhi asymptomatically in their gallbladder are thought to play an important role in the maintenance and dissemination of *S*. Typhi strains (*Dongol et al., 2012*), and *S*. Typhi is hypothesized to replicate extracellularly in the gallbladder (*Gonzalez-Escobedo et al., 2011*). Therefore, experimental systems that mimic the replication of *S*. Typhi in the human gallbladder need to be developed to determine if this environment preferentially favors the onward transmission of specific genotypes. To advance our understanding of the impact of these *gyrA/parC* mutations on *S*. Typhi epidemiology, we suggest validating our findings in more illustrative physiological systems, such as the *S*. Typhimurium biliary carriage model (*Crawford et al., 2010*) or a human *S*. Typhi challenge model (*Pollard et al., 2012*).

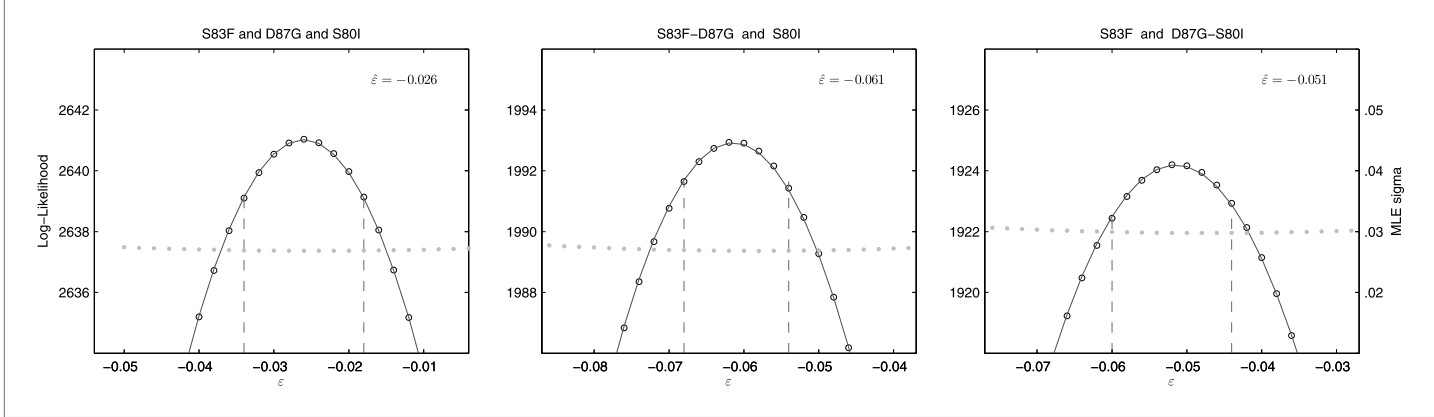

**Figure 6**. Likelihood profiles for the epistasis coefficient ($\hat{\varepsilon}$) of three possible epistatic interactions that could have generated the triple-mutant S83F-D87G-S80I. The interaction types are described on the top of each panel. The left panel shows the epistatic interaction among three single mutations. The middle and right panels show the epistatic interaction between a single mutation and a double mutation (joined by a hyphen). Open circles correspond to likelihood values; the filled gray circles correspond to the maximum likelihood estimate (MLE) for the variance parameter σ, describing the 24-hourly variance in both process and measurement. The MLE epistasis coefficient $\hat{\varepsilon}$ is shown in the top right of each panel. Vertical dashed lines demark the 95% confidence intervals for the MLE $\hat{\varepsilon}$.

Notwithstanding these experimental limitations, the positive selection coefficients we measure for our *gyrA* and *parC* mutations are consistent with the current understanding of the microbiology and molecular epidemiology of *S.* Typhi. The past two decades have seen the emergence of *S.* Typhi strains exhibiting reduced susceptibility to fluoroquinolones (*Wain et al., 1997*; *Parry et al., 1998*), resulting in the widespread distribution of these strains in almost all locations where typhoid is endemic. We now know that the majority of this epidemic is associated with one particular genotype (H58), which has swept across Asia and into Africa (*Holt et al., 2010*; *Kariuki et al., 2010*; *Holt et al., 2011*), displacing other genotypes in the process. A potentially crucial factor catalyzing the spread of this genotype is an S83F mutation in the *gyrA* gene. While this mutation is not unique to this genotype (*Roumagnac et al., 2006*), there is a strong association between the H58 strains and the S83F *gyrA* mutation. There are currently no data directly comparing *S.* Typhi genotype and disease outcome in typhoid patients; there is, however, a strong correlation between reduced susceptibility to fluoroquinolones (caused by the S83F mutation) and prolonged infection (*Parry et al., 2011*). Our data suggest that the dominance of the H58 genotype, the current global clonal expansion, and the emergence of other strains with reduced susceptibility to fluoroquinolones may not stem solely from therapeutic fluoroquinolone usage. Indeed, positive selection of bacteria with *gyrA* mutations has been predicted by mechanisms inducing DNA supercoiling continuation via suboptimal topoisomerases and modifications that enhance the functionality of the *gyrA* promoter (*Marcusson et al., 2009*; *Balsalobre et al., 2011*).

There are several outstanding questions regarding the impact of *gyrA* mutations on the evolution of *S.* Typhi that need to be addressed. (1) Why have certain high-fitness mutation combinations not arisen in natural populations of *S.* Typhi in the absence of fluoroquinolones? It may simply be improbable for an individual organism to acquire two specific mutations at the same time, especially given the slightly deleterious effects of D87A and D87G. However, this would not explain the absence, until recently, of S83F-D87N ($\hat{s} = +.074$) or of the S83F mutation itself ($\hat{s} = +.013$). Perhaps, for mechanistic reasons, these mutations are unlikely to emerge in the absence of fluoroquinolone pressure. (2) Why have *S.* Typhi strains with only certain *gyrA*/*parC* mutations been described in clinical typhoid? We know that the S83F mutation has emerged and spread recently; therefore, we suggest that strains containing one or two additional resistance mutations may not have had enough time to achieve high frequency in the *S.* Typhi population. In addition, there is a general lack of systematic cross-sectional analyses for *gyrA*/*parC* mutations in *S.* Typhi. Data from the published studies known to us suggest that, with the exception of S83F and S83Y, the clinical frequencies of the fluoroquinolone-resistant mutants described here are below 3% (*Roumagnac et al., 2006*; *Chau et al., 2007*; *Parry et al., 2010*; *Emary et al., 2012*). However,

because the majority of the bacterial isolates described in these studies are not recent, some of these genotypes may now be circulating at higher frequencies. (3) Why do S83F mutants dominate globally? Our results suggest that the S83F and the S80I mutants would be positively selected in nature in the absence of compensatory mutations or any selective pressure induced by antimicrobial usage. Of these two mutations, it is S83F that exhibits both the greatest selection coefficient and the highest MIC to the fluoroquinolones, making the emergence or establishment of S80I variants evolutionary less likely. The S83F mutation provides the most probable primary foundation for the evolution of fluoroquinolone-resistant *S*. Typhi. Phylogenetic investigations have shown that the S83F mutation has arisen on multiple occasions in several lineages (*Roumagnac et al., 2006*; *Holt et al., 2008*). Additionally, the ability of S83F mutants to catalyze positive epistatic interactions with other otherwise neutral mutations suggests that new S83F combinations may be observed more frequently in the future (*Koirala et al., 2012*).

Typhoid is a disease that necessitates antimicrobial therapy and the results presented here have repercussions for typhoid therapy and control. Currently, the control of typhoid across Asia and Africa relies on fluoroquinolone treatment (*WHO, 2003*), often prescribed in endemic settings for any non-specific low-grade febrile disease. However, many locations are observing a preponderance of strains that have elevated MICs to fluoroquinolones, resulting from the *gyrA* and *parC* mutations we have assessed here (*Holt et al., 2010*; *Kariuki et al., 2010*; *Holt et al., 2012*). These observations are of great concern as incremental increases in MICs to fluoroquinolones correspond directly with treatment failure and disease severity (*Parry et al., 2011*). Furthermore, antimicrobial resistance and an inability to clear infection may have obvious consequences for prolonged asymptomatic carriage of these pathogens. As the majority of *S*. Typhi strains described here do not exhibit a fitness cost, an antimicrobial control strategy, including the withdrawal of fluoroquinolones from general usage within the population, is unlikely to reduce the population-level frequencies of antimicrobial resistance to these drugs. In fact, our results indicate that there would be a continued rise in the frequency of fluoroquinolone-resistance, even in the absence of sustained drug exposure.

In conclusion, there is no other bacterial pathogen of which we are currently aware whose primary routes of drug resistance evolution are associated with such dramatic increases in intrinsic fitness. Our findings have important implications for the selection, maintenance, and treatment of drug-resistant members of the *Enterobacteriacea*, predicting that fluoroquinolone-resistant strains may be naturally maintained even if the use of this crucial group of antimicrobials is restricted.

## Materials and methods

### Bacterial strains, culture media and antimicrobial susceptibility testing

The attenuated *S*. Typhi Ty2 strain BRD948, containing deletions in the *aroA*, *aroC* and *htrA* genes, was the parent for all bacterial strains (*Tacket et al., 1997*). *S*. Typhi BRD948 is a well-characterized laboratory *S*. Typhi strain, with the attenuating mutations in the *aro* locus making it auxotrophic for aromatic compounds. These mutations affect the ability to grow in the intracellular compartment due to limitation of exogenous aromatic metabolites at this site. This strain is safe for use in a containment level two laboratory and avoids considerations associated with the genetic manipulation and the introduction of antimicrobial resistance mutations into an invasive human pathogen (*Bishop et al., 2008*).

All genetic manipulations were performed using Luria–Bertani (LB) media, all competition assays were performed using minimal (M9) media. As *S*. Typhi BRD948 is an aromatic auxotroph, growth media were supplemented with 40 mg/l each of l-phenylalanine and l-tryptophan, and 10 mg/l of p-aminobenzoic acid and 2,3-dihydroxybenzoic acid (*aro* mix). When required, media were supplemented with chloramphenicol, ampicillin, or nalidixic acid, at concentrations of 15, 50 and 20 mg/l, respectively. Fluoroquinolone susceptibility testing was performed by assessing the MICs for all strains against nalidixic acid, levofloxacin, ciprofloxacin, ofloxacin, and gatifloxacin by E-test on Mueller–Hinton media containing *aro* mix, following manufacturer's recommendations (AB Biodisk, Sweden).

### Construction and screening of *S*. Typhi mutants

All mutations were constructed by allelic exchange using derivatives of the suicide vector pJCB12 containing DNA fragments incorporating point mutation(s) of *gyrA*, *parC* and *aroC* constructed by

overlap extension PCR (PfuUltra DNA polymerase [Stratagene, La Jolla, USA]) (*Supplementary file 1*) (*Turner et al., 2006*). Fragments were ligated into pJCB12 using appropriate restriction enzyme sites (New England Biolabs Ltd, USA), propagated in *E. coli* CC118 λpir, and introduced into *S.* Typhi BRD948 by electrotransformation using 0.1 cm cuvettes in a GenePulser XcellTM (Bio-Rad Laboratories, Hemel Hempstead, UK) at 1.6–1.8 kV, 25 µF and 200 O. Plasmid DNA was prepared with the plasmid midi kit (QIAGEN, USA) and genomic DNA was prepared with the Wizard genomic DNA purification kit (Promega, USA) unless stated otherwise. Chloramphenicol-resistant transformants were screened by PCR (Platinum PCR Supermix' [Invitrogen, USA]) to identify successful recombinants. Recombinants were grown in the absence of antimicrobial selection to allow homologous recombination and derivatives were selected by growth in LB-broth containing chloramphenicol and ampicillin followed by plating on NaCl-free LB-agar supplemented with 5% sucrose. Appropriate colonies were selected and stored at −80°C until required. Mutants were screened for susceptibility to ciprofloxacin using antimicrobial discs or, for the *parC* and *aroC* mutants, by quantitative real-time PCR. The *gyrA*, *gyrB* and *parC* loci in all mutants were sequenced to ensure the correct nucleotide substitutions.

## Bacterial enumeration

Bacterial cultures were agitated and 100 µl aliquots were mixed with 900 µl of sterile phosphate buffered saline (PBS). Bacterial cultures were diluted, with thorough mixing, to $10^{-8}$ dilutions in 900 µl of sterile PBS. For enumeration, $3 \times 20$ µl of the $10^{-5}$, $10^{-6}$, $10^{-7}$, $10^{-8}$ dilutions were inoculated onto quadrants of LB *aro* mix plates with and without nalidixic acid, were appropriate. Plates were incubated overnight at 37°C and the dilution from each plate containing <20 colonies per 20 µl were counted; the median number of colonies was recorded. The number of bacterial generations was calculated from the number of cell divisions required to generate the stationary phase cfu/ml$^{-1}$ from the starting inoculum.

## Competitive growth assays

All experimental bacterial strains are shown in *Table 1*. Individual *S.* Typhi mutants were grown overnight on LB aro mix agar without antimicrobial supplementation. Single colonies were picked and inoculated into 10 ml of M9 aro mix broth and incubated overnight (16 hr +/− 2 hr) at 37°C with agitation. The bacterial growth at stationary phase was enumerated to ensure standardization in bacterial colony saturation. Through a series of growth dynamics experiments over 24-hr time periods we found that the parent *S.* Typhi strain and the mutants generated reproducible and comparable stationary phase bacterial counts. The concentration of bacterial cells was measured by OD600 and adjusted with M9 broth to match the parent *S.* Typhi strain, prior to mixing for competitive growth assays. 5 µl of the parent *S.* Typhi strain (approximately $1 \times 10^7$ organisms) was inoculated in 10 ml of M9 aro mix broth into an Erlenmeyer flask concurrently with the same concentration as a mutant. Both inocula were enumerated at the time of mixing to ensure accuracy (1:1 ratio). The bacterial mixture was incubated for a period of 24 hr (± 2 hr) at 37°C with circular agitation (speed 3.6, Lab companion SI-300 shaking incubator, South Korea). The following day, 50 µl of the stationary phase bacterial culture was removed for colony counting, 1 ml was stored at −80°C for DNA extraction and 10 µl was transferred into a secondary sterile Erlenmeyer flask containing 10 ml of M9 aro mix. The bacterial culture was incubated as before, and after a 24-hr growth period, identical volumes were removed and processed as before. Each experiment continued for a period of 15 days.

## Calculation of allele frequencies by pyrosequencing

Bacterial cells from each time point were thawed and DNA was extracted using heat treatment and phenol-chloroform purification. Briefly, a 100 µl aliquot of the concluding bacterial culture was agitated and incubated at 100°C for 10 min before returning to ambient temperature. The resulting solution was centrifuged and an equal volume of phenol-chloroform was added. The mixture was vortexed and centrifuged at 13,200 rpm for 30 min in a benchtop microfuge (Eppendorf, USA). The aqueous layer was removed, placed in a sterile microfuge tube and prepared immediately for pyrosequencing.

The DNA from the competitive growth assays was PCR amplified (Platinum PCR Supermix [Invitrogen, USA]) in triplicate using biotinylated primer pairs targeting the region containing the SNP distinguishing the two organisms in the assay, that is mutations in *gyrA*, *parC* and *aroC* (*Supplementary file 1*).

PCR amplifications were performed in 60 µl reactions containing 1 × NH$_4$ buffer, 1.5 mM of MgCl$_2$, 200 µM of dNTP, 10 pM of each primer, 1.25 Units of Hotstart DNA polymerase (Qiagen, USA) and 5 µl of template DNA. Reactions were cycled once at 95°C for 15 min, followed by 30 cycles of 94°C for 1 min, 55°C for 1 min and 72°C for 1 min, with a final elongation of 72°C for 5 min. All PCR amplifications were visualized on 1% agarose gels prior to pyrosequencing.

A pyromark Q96 ID DNA pyrosequencer (Biotage, Sweden) was used to detect the proportion of each allele in the competitive assays at each time point as per the manufacturer's recommendations. PCR amplicons were combined with 56 µl of binding buffer and 4 µl of streptavidin sepharose beads. The resulting mixture was agitated for 5 min before denaturation in denaturation buffer and washing with the Vacuum Prep Tool (Biotage, Sweden). DNA fragments were transferred into a 96-well plate containing 3.5 pmol of sequencing primer in 40 µl of annealing buffer and the DNA sequencing reaction was performed using the Pyro Gold Kit (Biotage, Sweden). The ratio of the parent *S.* Typhi strain to the engineered mutant was determined by the ratio of the non-wild-type allele to wild-type allele at the known SNP position. The allelic quantification mode in the software PyromarkID v1.0 (Biotage, Sweden) was used to quantify the proportion of each allele at each time point.

## Validation of allele frequency measurements

The accuracy of allele frequency measurement by the pyrosequencer was validated by comparing the predicted value of the S83F *gyrA* mutation and the parent strain. Single colonies of *S.* Typhi BRD948 and the engineered S83F mutant were cultured separately in 10 ml of M9 *aro*, overnight at 37°C with agitation. Bacterial concentrations (OD$_{600}$) were determined and adjusted to the same value with M9 *aro*. These cultures were then combined in ratios of 0:10, 1:9, 2:8, 3:7, 4:6, 5:5, 6:4, 7:3, 8:2, 9:1 and 10:0 in a total volume of 200 µl. DNA was extracted as before from 100 µl of the mixed cultures. DNA was subjected to pyrosequencing and compared to the expected frequency of the mutant, as before. The bacterial mixtures were additionally enumerated by colony counting and compared to the expected frequency of the mutant and the pyrosequencing data. These experiments were replicated 18 times. The results were compared by linear regression and F-test to compare variances; statistical analyses were performed in MATLAB (Mathworks, Natick, MA, USA).

## Maximum likelihood estimate of selection coefficients and epistasis parameters

For each strain, five independent competition assays were performed over a 15-day period. The competition period was modeled and fit with a standard Wright-Fisher model consisting of a wild type of unit fitness ($w = 1$) and a mutant with fitness equal to $w = 1 + s$, using the allelic frequencies generated by SNP-specific pyrosequencing. In this type of model, fitness differences are expressed on a per-generation basis, meaning that during one generation of bacterial replication a mutant strain is expected, on average, to generate $1 + s$ offspring for every one offspring generated by the wild-type (parent) strain; if the estimate of $s$ is negative, the mutant is less fit than the wild type and generates, on average, fewer surviving offspring than the wild type. Using $y_t$ to denote the measured frequency of an allele at time $t$, and $z_t$ to denote the true allele frequency at time $t$, the likelihood function for a single 15-day competition experiment was defined as:

$$\prod_{t=1}^{15}\int_0^1 f\left(y_{t+1}|z_t,\sigma_1\right).g\left(z_t|y_t,\sigma_2\right)dz_t,$$

where $g$ is the probability density function describing the combined measurement and sampling error during the pyrosequencing procedure, and $f$ is the probability density describing the process error from time $t$ to time $t + 1$ and the measurement error at time $t + 1$. Because the probability of measurement error at time $t + 1$ will be included in the next term in the product—when we factor in the likelihood of observation $y_{t+2}$ conditioned on $y_{t+1}$—we can simply view $f$ as describing process variation. Both the density functions $f$ and $g$ were modeled as normal distributions, truncated outside the closed interval [0, 1], and renormalized to integrate to unity. Because the allele frequencies did not approach within one standard deviation of the frequency boundaries zero and one, a normal distribution was an appropriate approximation of the binomial Wright–Fisher process for the density function $f$. As all of the allele frequency trajectories were quite regular, we set $\sigma_1 = \sigma_2$ in order to have a single variance parameter for the system describing the variation in allele frequency introduced in a 24-hr period. Likelihoods values were multiplied across the five replicates. In the likelihood expression above, the

density function $f$ depends on the fitness coefficient $s$ via the mean of this normal distribution, which is $(1 + s) z_t/(1 + sz_t)$. Likelihood optimization was performed with a standard Nelder–Mead method (C++ with GSL Library, http://gnu.org/software/gsl), and confidence intervals were obtained using likelihood profiles (*Figures 2, 4 and 6*).

For the analysis of epistasis parameters, the same likelihood equation was used, with the fitness coefficient of the double mutant defined as $w_{ij} = (1 + s_i)(1 + s_j) + \varepsilon_{ij}$; the parameters $s_i$ and $s_j$ are the selection coefficients for strains with mutations $i$ and $j$, respectively, and $\varepsilon_{ij}$ is the multiplicative epistasis parameter for the strain containing both mutations $i$ and $j$. In this case, likelihoods were computed across 15 replicates—five replicates each of the two strains with single mutations, and five replicates of the strain with both mutations. The corresponding epistasis equation for a triple mutant is $w_{ijk} = (1 + s_i)(1 + s_j)(1 + s_k) + \varepsilon_{ijk}$; and in this case the epistatic interaction is defined as the fitness interaction among all three mutations. For the triple mutant, the epistatic interaction can also be modeled as the fitness interaction between a single new mutation emerging onto a genetic background already containing two mutations. In this case, the fitness is written down as $w_{ijk} = (1 + s_{ij})(1 + s_k) + \varepsilon_{(ij)(k)}$. Both types of epistatic interactions for the triple mutant were considered, and estimates of the epistasis parameter are shown in *Figure 6*.

## Acknowledgements

This work was supported by The Wellcome Trust of Great Britain (089276/B/09/Z). The strains in this study were generated with the support of Wellcome Trust grant 098051. The authors wish to acknowledge the efforts of Ms Song Chau and others members of the microbiology laboratory at OUCRU. SB and MFB are holders of Sir Henry Dale Fellowships, jointly funded by the Wellcome Trust and the Royal Society (100087/Z/12/Z and 098511/Z/12/Z, respectively).

## Additional information

### Competing interests

JF: Jeremy Farrar is Director of the Wellcome Trust, one of the three founding funders of *eLife*. The other authors declare that no competing interests exist.

### Funding

| Funder | Grant reference number | Author |
|---|---|---|
| Wellcome Trust | 089276/B/09/Z | Pham Thanh Duy, Tran Vu Thieu Nga, Tran Thi Ngoc Dung, Voong Vinh Phat, Tran Thuy Chau, Jeremy Farrar, Maciej F Boni, Stephen Baker |
| Sir Henry Dale Fellowship, jointly funded by the Wellcome Trust and the Royal Society | 100087/Z/12/Z | Stephen Baker |
| Sir Henry Dale Fellowship, jointly funded by the Wellcome Trust and the Royal Society | 098511/Z/12/Z | Maciej F Boni |
| Wellcome Trust | 098051 | A Keith Turner |

The funders had no role in study design, data collection and interpretation, or the decision to submit the work for publication.

### Author contributions

PTD, Performed the experiments, Conception and design, Analysis and interpretation of data; TVTN, Performed the experiments, Conception and design; TTND, Performed the experiments; VVP, Performed the experiments; TTC, Generated the strains used in the experiments; AKT, Generated the strains used in the experiments; JF, Conception and design; MFB, SB, Conception and design, Analysis and interpretation of data, Drafting or revising the article

## Additional files

### Supplementary files
• Supplementary file 1. Oligonucleotides used in this study.

---

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
