## [Decision Letter]

Thank you for sending your work entitled “Fitness benefits in fluoroquinolone-resistant *Salmonella* Typhi in the absence of antimicrobial pressure” for consideration at *eLife*. Your article has been favorably evaluated by Prabhat Jha and 3 reviewers, one of whom, Sebastian Gagneux, has agreed to reveal his identity.

The Senior editor and the three reviewers discussed their comments before we reached this decision, and the Senior editor has assembled the following comments to help you prepare a revised submission.

Major comments:

1) These are potentially novel and indeed counter-intuitive findings given what we know about antimicrobial-induced resistance. Given their potential importance, it is important that you revise the Discussion and conclusions to avoid making statements that are too strong. As noted below, the results do require replication and their implication in vivo or in clinical settings is not yet clear. Thus, we suggest you examine the manuscript very carefully to tone down claims and to nuance any external generalizations outside your experimental model.

2) The manuscript does not address four specific limitations or potential future work, and would be strengthened by directly discussing these aspects (mostly in the Discussion, or where relevant in the Results).

A) The strain of *S.* typhi you used was an *aro* mutant, which is relevant for laboratory analysis; however, mutations in the *aro* system markedly attenuate the organism and make it unable to survive intra-cellularly, which perhaps is the main ecological niche for *S.* typhi. As such, it is possible that the results reported here may not hold true for wild type *aro* strains in vitro and, perhaps most importantly, may not be true in vivo.

B) With regard to the in vitro model, the use of M9 *aro*-supplemented media is appropriate for the model that was used, but, once again, the model can only go so far. The results would be most pertinent to survival and environmental reservoirs, and, as such, do have some significance, since *S.* typhi survives in environmental and municipal water systems. However, the results don't predict anything that may be happening inside of macrophages, enterocytes, or the biliary system. To perform that work is beyond the scope of the current manuscript, but is perhaps the most important work to be performed. Please comment on how the work could be repeated using ex vivo macrophage model systems with wild type *S.* typhi as the backbone strain. Similarly, intestinal epithelial tissue culture systems could be used.

C) It would be of interest also to know if similar mutations in *S.* Typhimurium confer similar in vitro fitness benefit, since that could open the possibility of more facility for studying these interesting biological observations in the mouse model.

D) The epistasis coefficients of double mutant strains were also calculated. It was not understood why epistasis coefficient of triple mutant strain was not determined.

---

## [Author Response]

*1) These are potentially novel and indeed counter-intuitive findings given what we know about antimicrobial-induced resistance. Given their potential importance, it is important that you revise the Discussion and conclusions to avoid making statements that are too strong. As noted below, the results do require replication and their implication in vivo or in clinical settings is not yet clear. Thus, we suggest you examine the manuscript very carefully to tone down claims and to nuance any external generalizations outside your experimental model*.

In order to include the necessary caveats about linking our in vitro results with how these fitness differences would be manifested in an epidemiological setting, we have added a new section to the Discussion.

Paragraphs 3, 4, and 5 of the Discussion describe the limitations of the current in vitro and ex vivo models of Typhi replication, as well as results on *Salmonella* Typhimurium fitness estimates in a murine challenge model that showed reduced fitness of fluoroquinolone-resistant strains. The limitations of this animal model are that both the pathogen and host differ from *Salmonella* Typhi infections in humans, and that mice typically succumb to infection within seven days limiting the amount of data available in the competition experiment. Paragraph 5 of the Discussion explicitly addresses the challenges of associating fitnesses measured in any experimental system to those that would be observed in the natural epidemiological environment where humans are infected with Typhi. We discuss a relevant but unstudied feature of Typhi transmission (gall bladder carriage) and suggest two experimental systems that may be most appropriate for future studies aiming to assess transmission/fitness differences in Typhi.

We have changed the tone/wording in several other parts of the manuscript, and we feel that the current general tone of the manuscript is appropriate in alerting the reader that translation of results from experimental systems to epidemiological contexts is filled with uncertainty. If the editors/referees had some specific sections of our manuscript in mind, we would be happy to consider revising those as well.

*2) The manuscript does not address four specific limitations or potential future work, and would be strengthened by directly discussing these aspects (mostly in the Discussion, or where relevant in the Results)*.

*A) The strain of* S. *typhi you used was an* aro *mutant, which is relevant for laboratory analysis; however, mutations in the* aro *system markedly attenuate the organism and make it unable to survive intra-cellularly, which perhaps is the main ecological niche for* S. *typhi. As such, it is possible that the results reported here may not hold true for wild type* aro *strains in vitro and, perhaps most importantly, may not be true* in vivo.

As stated by the referees, the major limitation of this work was the use of an *S.* Typhi *aro* mutant strain in an in vitro competitive growth assay. This strain was selected to avoid potential biological safety issues of introducing known antimicrobial resistance mutations into an invasive human pathogen. As *S.* Typhi is a human-restricted facultative intracellular organism, our findings on fitness differences may not directly reflect the evolution/competition of strains during an acute Typhi infection. However, we wish to highlight that *Salmonella*, including *S.* Typhi, can survive for prolonged periods in water supplies and the gallbladder of asymptomatic carriers (Baker et al. 2011, Mermin et al. 1999, [12]). Therefore, our findings need to be taken in context with the current understanding of *Salmonella* biology. These results require additional experimentation, with non-attenuated strains, to provide further evidence on the selective potential of these mutations in nature.

These points are now highlighted in paragraphs 3–5 of the Discussion section. We acknowledge the limitations of the current Typhi experimental systems (in vitro, ex vivo, and mouse models) and we suggest which of these models may be the most appropriate in validating the fitness estimates we measured in our in vitro assays.

*B) With regard to the in vitro model, the use of M9* aro*-supplemented media is appropriate for the model that was used, but, once again, the model can only go so far. The results would be most pertinent to survival and environmental reservoirs, and, as such, do have some significance, since* S. *typhi survives in environmental and municipal water systems. However, the results don't predict anything that may be happening inside of macrophages, enterocytes, or the biliary system. To perform that work is beyond the scope of the current manuscript, but is perhaps the most important work to be performed. Please comment on how the work could be repeated using ex vivo macrophage model systems with wild type* S. *typhi as the backbone strain. Similarly, intestinal epithelial tissue culture systems could be used*.

This is an excellent point and perhaps the most challenging step in translating experimental results to natural epidemiological settings.

We have addressed this comment in paragraphs 4 and 5 of the Discussion, where we discuss the benefits and limitations of the epithelial cell and macrophage models as well as the classical *S.* Typhimurium challenge model.

In paragraph 5, we outline some consequences of *S.* Typhi survival in water systems and *S.* Typhi survival and extracellular replication in the human gall bladder. Extracellular experimental systems may be important in understanding and predicting the evolution of certain Typhi genotypes, and the biliary carriage model (Typhimurium) and human challenge models may be the most appropriate systems in which to assess the competitive interactions among genotypes.

*C) It would be of interest also to know if similar mutations in* S. *Typhimurium confer similar in vitro fitness benefit, since that could open the possibility of more facility for studying these interesting biological observations in the mouse model*.

We think that this would be the next sensible set of experiments to perform. Observing the same phenotype in an in vivo system would add substantial support to these mutants playing a role in the transmission and epidemiology of typhoid. The Typhimurium model would permit a degree of experimental flexibility in understanding these mutations in greater detail, but may not precisely reflect the effects seen during acute typhoid fever in humans. There is an extensive literature on Typhimurium infections in animal models and the various pros and cons of this experimental system. To determine if there were any existing data regarding these Salmonella mutations in animal models we performed an extensive literature search over “fluoroquinolone”, “Typhimurium”, “gyrA”, and certain specific mutations of interest (over 400 studies met parts of these criteria). Only two studies assessed fitnesses of FQ-resistance mutations or *gyrA* mutations relevant to our study (19; 16). These studies did not find fitness benefits associated with *gyrA* mutations, although the high-fitness mutants we found were not assessed in these two studies. In paragraph 4 of the Discussion, we cite these two papers and discuss some of the limitations of the Typhimurium murine model.

*D) The epistasis coefficients of double mutant strains were also calculated. It was not understood why epistasis coefficient of triple mutant strain was not determined*.

In the revision, we have added calculations for epistasis coefficients of the triple mutant. We avoided this initially, because there is no way to uniquely identify an interaction among three mutations. The epistasis coefficient of a triple mutant could be written down as a parameter describing deviation from “independence of fitness effects” for three individual mutations. Or, the epistasis coefficient could be written down as the interaction between a single new mutation and a double mutant; there are three different ways to write down this interaction.

We could only compute the epistasis coefficient for 3 out of these 4 possibilities, as the S83F-S80I double mutant was not generated for our experiments.

We have added a description to the Materials and methods section showing how these three epistasic interactions for the triple mutants were computed, and we show the MLE estimates and confidence intervals in Figure 6. The epistatic interaction for the triple mutant is antagonistic, whether it is computed as an interaction among three mutations or an interaction between a double mutant and a single mutant.